# Dietary Inflammatory Index and Its Association with the Prevalence of Coronary Heart Disease among 45,306 US Adults

**DOI:** 10.3390/nu14214553

**Published:** 2022-10-28

**Authors:** Lida Wu, Yi Shi, Chaohua Kong, Junxia Zhang, Shaoliang Chen

**Affiliations:** Department of Cardiology, Nanjing First Hospital Affiliated to Nanjing Medical University, Nanjing 210006, China

**Keywords:** Dietary Inflammatory Index, coronary heart disease, NHANES, restricted cubic spline, cross-sectional study

## Abstract

Inflammation plays a pivotal in the occurrence and development of coronary heart disease (CHD). We aim to investigate the association between the Dietary Inflammatory Index (DII) and CHD in the present study. In this cross-sectional study, adult participants from the National Health and Nutrition Examination Survey (NHANES) (1999–2018) were enrolled. The social demographic information, lifestyle factors, blood biochemical measurements, dietary information, and CHD status of all the participants were systematically collected. Multivariable logistic regression was adopted to investigate the association between the risk of CHD and the DII. Besides, restricted cubic spline (RCS) analysis was used to explore whether there was a nonlinear association of the DII and CHD. Subgroup analysis stratified by sex, age, race/ethnicity, and BMI was conducted to evaluate the association of the DII and CHD among different populations. A total of 45,306 adults from NHANES (1999–2018) were included. Compared with individuals without CHD, the DIIs of the participants with CHD were significantly elevated. A positive association was observed between the DII and CHD in multivariable logistic analysis after adjusting for age, sex, race/ethnicity, education levels, smoking, drinking, diabetes, hypertension, and body mass index (BMI). Results of RCS analysis suggested a nonlinear relationship between the DII and CHD. In addition, the increment of the DII had a greater impact on female individuals compared with male individuals. The DII is closely associated with the risk of CHD. For better prevention and treatment of CHD, more attention should be paid to controlling dietary inflammation.

## 1. Introduction

Coronary heart disease (CHD) is a chronic complex disease with high morbidity and mortality, mainly caused by atherosclerotic lesions in coronary vessels, eventually leading to myocardial infarction (MI) and stroke [1]. Coronary atherosclerosis originates from the intima of coronary arteries, and its pathological process goes through the early stage of endothelial dysfunction and lipid deposition, the middle stage of atherosclerotic plaque and fibrous plaque, and the late stage of composite plaque, such as calcium salt deposition and the formation of calcified plaque. Moreover, lesion stability is closely related to the occurrence of MI. Vulnerable plaques refer to those that rupture easily and are unstable and prothrombotic. Inflammation plays an important role in the formation of vulnerable plaques and plaques rupturing, which can trigger the formation of a blood clot, eventually leading to MI [2]. There are plenty of risk factors for CHD, including dyslipidemia, hypertension, insulin resistance, hypercoagulability, and inflammatory responses [3,4,5]. Among all risk factors, considerable attention has been paid to the roles of inflammatory responses in the development of CHD [6,7].

CHD is closely associated with unhealthy eating habits, which can be achieved through the influence of chronic inflammation. Inflammation is of great importance for the occurrence and development of CHD. Numerous researchers have investigated the association between eating habits and CHD risk. High-fat diet is one of the key factors that induce inflammation. Besides, obesity caused by a high-fat diet is often accompanied by a series of chronic inflammatory diseases, such as type 2 diabetes and hypertension, eventually leading to CHD [8,9]. Consuming saturated fatty acids (SFA) substantially increases serum cholesterol levels and causes inflammation, while consuming polyunsaturated fatty acids (PUFA) decreases them. It has been reported that linolenic acid (ALA), linoleic acid (LA), docosahexaenoic acid (DHA), eicosapentaenoic acid (EPA), n−3 fatty acids, and n−6 fatty acids collectively have a protective effect [10]. Besides, vitamins and other trace elements are also important to systemic inflammation and CHD. For example, vitamins D and A has anti-inflammatory properties. Moreover, zinc is important for a variety of enzymes in cells and reduces the damage caused by inflammation [11].

Considering inflammation as a fundamental component of atherosclerosis, measuring inflammation levels is likely to be useful for predicting and protecting CHD. The Dietary Inflammatory Index (DII) was first proposed by Shivappa et al. in 2014 based on the summary of published literatures [12]. The DII is a novel, validated, and comprehensive tool for quantifying the proinflammatory or anti-inflammatory potential of the diet. It was assessed by calculating the inflammatory index for each dietary component consumed and, finally, integrating individual dietary components. The DII has already been associated with blood inflammatory measurements, including interleukin (IL)-1β, IL-6, IL-10, tumor necrosis factor (TNF)-α, and C-reactive protein (CRP) and widely used to assess the overall proinflammatory and anti-inflammatory properties of an individual’s diet [13]. 

Nevertheless, to our knowledge, relatively few researchers have investigated the association between the DII and CHD. Therefore, we conducted this cross-sectional study to explore the association between the DII and the prevalence of CHD in a large multiracial cohort in the US, which may be helpful for the prevention and treatment of CHD. 

## 2. Materials and Methods

### 2.1. Study Design and Participants

The National Health and Nutrition Examination Survey (NHANES) is a program designed to measure the health and nutrition status of adults and children in the US. A mass of data in the NHANES database has been analyzed extensively, which is of great help in unraveling the etiologies, understanding the epidemiology, and searching for novel biomarkers of different diseases [14,15,16]. The method of “stratified multistage probability sampling” was adopted to screen out representative participants in the survey. Detailed methods are described in the NHANES website (http://www.cdc.gov/nchs/nhanes.htm (accessed on 1 August 2022)). All participants enrolled in NHANES provided written informed consent, and the whole procedures were approved by the Institutional Review Board of the Centers for Disease Control and Prevention. An analysis of 10 consecutive NHANES circles from 1999/2000 to 2017/2018 was conducted in the present study. The exclusion criteria were as follows: (1) participants’ age < 18 or ≥ 80 years, (2) participants with missing CHD status, (3) participants with no DII, or missing dietary information (Figure 1).

### 2.2. Dietary Information

The Nutrition Methodology Working Group of the survey collected all of the dietary information in the mobile examination center (MCE) through 24 h dietary recall interviews. We calculated the DII based on 28 dietary components and following the protocol reported by Shivappa et al. [12]. There were six different markers used to evaluate inflammation levels. If a dietary component can significantly increase the concentrations of interleukin (IL)-1β, IL-6, tumor necrosis factor-α (TNF-α), and C-reactive protein (CRP) or reduce the concentration of IL-4 and IL-10, a score of “+1” was assigned; on the contrary, a score of “−1” was assigned. If the dietary component did not change inflammatory markers, it was considered to have no inflammatory characteristics, and a score of “0” was assigned. In the overall inflammation index, the positive value represents the proinflammatory potential of the diet, the negative value represents the anti-inflammatory ability of the diet, and 0 means neither proinflammatory nor anti-inflammatory potential. If the diet contains more proinflammatory components, such as saturated fatty acids and carbohydrates, the DII is higher. On the contrary, the more anti-inflammatory ingredients, such as green vegetables, fruits, whole grains, and marine fish, the lower the DII. The association between the DII and systemic inflammatory level has already been validated. Studies have demonstrated that high-DII diet can increase IL-1β, IL-6, IL-10, TNF-α, and CRP levels in the human body, and the DII is now widely used to assess the overall inflammatory properties of diet [17]. It has been reported in previous studies that using less than 30 food ingredients could make sure the stable predictive ability of the DII [18]. We first analyzed the DII as a continuous variable, and then we equally classified the participants into four groups according to DII distribution: low DII (Q1), lower middle DII (Q2), higher middle DII (Q3), and high DII (Q4). The Healthy Eating Index (HEI) is a widely used dietary measurement designed by the United States Department of Agriculture to evaluate the quality of diet by comparing the intake of 13 components of one’s daily diet with the Dietary Guidelines for Americans. HEI-2015, the latest version of HEI, was also calculated in the present study according to the guideline. It ranges from 0 to 100, The higher HEI scores, the better quality of diet [19,20]. HEI was also first analyzed as a continuous variable, and then classified as quartiles.

### 2.3. Definition of CHD

Similar to previous published articles based on NHANES, CHD was mainly adjudicated by the history of CHD [21]. In the health questionnaires, participants were asked “whether a doctor or other health professional has ever told you that you had CHD”, and individuals were regarded as patients with CHD if the answer was “yes”.

### 2.4. Covariates

Age, race/ethnicity, and education levels were obtained from the demographic questionnaires. Diabetes history, alcohol consumption, and smoking status were also adopted from the health questionnaires. Race/ethnicity was classified as non-Hispanic white, non-Hispanic black, Mexican American, other Hispanic, and others. Education was categorized into three levels: below high school, high school, and above high school. After at least 8 h of an overnight fast, blood samples were collected and used to examine the levels of TG, total cholesterol (TC), low-density lipoprotein cholesterol (LDL-C), high-density lipoprotein cholesterol (HDL-C), red blood cells (RBC), white blood cells (WBC), platelet (PLT), neutrophil (NE), lymphocyte (LY), hemoglobin, glycosylated hemoglobin (HbA1c), and fasting blood glucose (FBG). The estimated glomerular filtration rate (eGFR) was calculated according to the Chronic Kidney Disease Epidemiology Collaboration creatinine equation. The NHANES website provided the detailed procedures in collecting blood biochemical measurements [22].

### 2.5. Statistical Methods

Continuous variables were presented as the mean ± standard deviation (SD) (normal distribution) or the median (interquartile range) (skewed distribution). We compared baseline characteristics among individuals with and without CHD based on independent t tests, chi-square test, and Mann––Whitney U test. Multivariable logistic regression was adopted to investigate the relationship between the risk of CHD and DII after adjusting for confounding factors (age, sex, race/ethnicity, education levels, smoking, drinking, hypertension, diabetes, and eGFR). Restricted cubic spline (RCS) analysis (with three piecewise points) was used to evaluate the nonlinear associations between the DII and CHD risk. Obesity was defined as BMI ≥ 30.0 kg/m^2^. Subgroup analysis, stratified by sex, age, race/ethnicity, and BMI, was conducted to evaluate the heterogeneity among different populations. The correlation between the DII and HEI was investigated using “Sperman” method. We also conducted multivariable logistic regression including both the DII and HEI to explore the association between the HEI and CHD risk. A two-sided *p*-value < 0.05 was considered significant. All statistical analyses were conducted using the R software (R Core Team, 2022, Vienna, Austria; version 4.1.6).

## 3. Results

### 3.1. Characteristics of the Study Population

A total of 45,306 participants from NHANES (1999–2018) were enrolled in the present study (Figure 1), of whom 1575 (3.47%) had CHD. Among all the participants, 21,857 (48.2%) were male, 19,322 (42.6%) were non-Hispanic white, and the median age of all the participants was 47 years. The detailed baseline characteristics of the participants grouped by CHD status are shown in Table 1. We observed a significant difference in the demographic and baseline clinical characteristics between individuals with CHD and those without CHD. Compared with participants without CHD, those with CHD were older, more often male (68.3% vs. 47.5%) and with lower educational levels. The prevalences of HBP, DM, angina, heart attack, HF, and stroke were all higher among participants with CHD compared with those without CHD. Moreover, the eGFR of the participants with CHD were significantly lower compared with those without hypertension (75.77 (60.20, 89.05) vs. 98.17 (82.95, 113.05)). Of note, the participants with CHD had a significantly higher DII (1.96 (0.40, 3.13) vs. 1.73 (0.21, 2.93)) compared with the participants without the DII. There is a higher proportion of individuals with high DII or higher middle DII in participants with CHD. We also explored the factors that contributed to the difference in the DII through comparing all of the components of the DII among the two groups. We found that individuals with CHD had higher inflammatory scores in fiber, MUFA, PUFA, niacin, vitamin A, thiamin, riboflavin, vitamin B6, vitamin C, vitamin D, vitamin E, magnesium, zinc, selenium, folic acid, β-carotene, alcohol, N6 fatty acids, and N3 fatty acids, but lower inflammatory scores in carbohydrate, protein, total fat, cholesterol, saturated fat, vitamin B12, iron, and energy (Table 2). Appendix A show the baseline characteristics and each component of the DII of the enrolled participants grouped by sex. Overall, male individuals presented a higher DII (2.12 (0.69, 3.1) vs. 1.29 (−0.19, 2.59)) and were more likely to develop CHD (3.8% vs. 1.7%).

### 3.2. Association of DII and Prevalence of CHD

Figure 2A shows the distribution of the DII among all the participants. As a continuous variable, a positive association was observed between the DII and incidence of CHD, with an OR of 1.09 (95% CI: 1.05–1.14) in the unadjusted logistic regression model. After adjusting for confounding factors, including age, sex, race/ethnicity, education levels, smoking, drinking, diabetes, hypertension, and BMI, the DII was still significantly associated with CHD in the fully adjusted model II (OR: 1.08; 95% CI: 1.03–1.14) (Table 3). Besides, RCS analysis was adopted in this study to further investigate the association between the DII and CHD. Results of RCS analysis also revealed that the DII was positively correlated with the incidence of CHD and in a nonlinear pattern (*p* for nonlinear = 0.023). The risk of CHD increased rapidly with the increase in DII, especially if the DII was greater than 2 (Figure 2). We also carried out a subgroup analysis stratified by sex and found that there was a sex difference between male and female individuals in the association of the DII and CHD. When the DII > 2, the risk of CHD increased more rapidly in female than in male participants (Figure 2B,C). Considering the nonlinear association pattern, all the participants enrolled were grouped according to the quartile of the DII. When treated as a categorical variable, individuals in the third quartile (OR: 1.15, 95% CI: 1.01–1.31) and fourth quartile (OR: 1.24, 95% CI: 1.09–1.41) of the DII had a higher risk of CHD compared with those with a low DII (first quartile) (Table 3).

### 3.3. Subgroup Analysis of DII and the Risk of CHD

Subgroup analysis stratified by sex, age, race/ethnicity, and BMI was carried out in this study to further investigate the relationship between the DII and CHD among different populations. As shown in Figure 3A, a positive association was found between the DII and CHD, and we highlighted this association and its degree among different populations. Moreover, the results of subgroup analysis suggested that the conclusion in this study is very stable. Consistent with the results of RCS analysis, the results of multivariable logistic regression also showed that the increment of the DII had a greater impact on female individuals compared with male ones (*p* for interaction = 0.031) (Figure 3B). However, there was no statistical difference between subgroups stratified by age (*p* for interaction = 0.11) (Figure 3C). We also found that the non-Hispanic black was more sensitive to the DII than the non-Hispanic white and other populations (*p* for interaction = 0.012) (Figure 3D). In addition, compared with individuals with obesity, the increment in the DII was more closely associated with the risk of CHD among those with normal weight (*p* for interaction = 0.032) (Figure 3E).

### 3.4. Association of DII and HEI

We also calculated another dietary index in this study (HEI), which is often used in evaluating the quality of diet in previous studies. We explored the correlation between the DII and HEI in the same population, as shown in Figure 4. The DII had a significant negative association with the HEI (R = −0.48; *p* < 0.001). However, results of multivariable logistic analysis including both the DII and HEI demonstrated that, after adjusting for confounding factors, the DII had a close association with CHD, but the association between the HEI and CHD was not observed in either the continuous or category variable (Table 4). The DII may be a better predictor of CHD given the focus on inflammation by the DII.

## 4. Discussion

It is gradually recognized that chronic inflammation plays a critical role in diverse pathological states and chronic diseases. Inflammation–endothelial dysfunction interaction initiates and propitiates atherosclerosis. The DII is a literature-derived tool that measures the inflammatory potential of a person’s diet. In this cross-sectional study, we explored the relationship between the DII and the prevalence of CHD based on a large population from NHANES. The main findings are as follows: (1) The DIIs in participants with CHD were significantly higher compared with those without CHD; (2) the DII had a positive association with the prevalence of CHD. The association of the DII and CHD was in a nonlinear pattern; and (3) compared with males, females were more sensitive to the DII. Moreover, the increment in the DII was more closely associated with the risk of CHD among individuals with normal weight and in the non-Hispanic black population. However, we can only draw correlation conclusions, considering that this study is a cross-sectional study. More prospective studies are needed to further investigate the association between the DII and CHD.

Unhealthy eating habits lead to chronic systemic inflammation, which is one of the most important characteristics of metabolic diseases [23]. Proinflammatory diets can increase the levels of inflammatory cytokines by promoting oxidative stress and immune disorders. Macrophages play an important role in this process through producing signals as free radicals, chemokines, and cytokines [24]. A healthy diet, such as the Mediterranean diet (rich in fruits and vegetables), is generally associated with lower levels of inflammatory markers, and a Western diet (rich in saturated fat and simple carbohydrates) is associated with higher levels of inflammatory markers [25]. Studies have already demonstrated that Western diet habits can cause hyperglycemia and hyperlipidemia and generate reactive oxygen species (ROS) through nonenzymatic glycosylation and glucose-induced NADH [26]. Western diet can also reprogram the intestinal microbial ecosystem and promote chronic inflammation [27]. Ketogenic diet, characterized by high fat, very low carbohydrate, and moderate protein, has been used in the treatment of obesity, diabetes, and Parkinson’s disease. Ketogenic diet mimics starvation and causes starvation-induced ketosis [28]. Some investigators thought ketogenic diet has an anti-inflammatory effect; however, it is partial due to the momentary and rapid weight reduction of these low-carbohydrate diets [29]. A recent study compared the DII and HEI across a variety of popular diet trends and fads and found that keto diet had a slightly proinflammatory tendency [30]. The DII, a literature-derived index to evaluate the inflammatory potential of an individual’s diet, has been widely used in recent years [18]. Results from a case–control study in South Korea revealed a close association between the DII and the risk of cervical cancer. Higher intake of anti-inflammatory dietary factors (fruits lower in sugar and plant foods rich in fiber) and reducing the consumption of proinflammatory factors (fried foods or processed foods high in saturated or trans fats) can reduce the risk of cervical cancer [31]. In another retrospective study, higher DII scores were associated with an increased risk of prostate cancer, and this effect was more pronounced in obese individuals [32]. Besides, a recent meta-analysis confirmed that elevated levels of the DII were associated with a 31% increased risk of depression, and the investigators also found a greater association between proinflammatory diet and depression in female individuals compared with male individuals [33].

Nevertheless, the association between the DII and CHD has not been reported yet. In the current study, our results based on 45,306 adults from NHANES showed significant increases in DII levels of patients with CHD. Inflammation plays a crucial role in atherosclerosis, which is one of the pivotal mechanisms of CHD [34]. We found that participants with CHD had higher inflammatory scores in dietary fiber, MUFA, PUFA, niacin, vitamin A, thiamin, riboflavin, vitamin B6, vitamin C, vitamin D, vitamin E, magnesium, zinc, selenium, folic acid, β-carotene, alcohol, N6 fatty acids, and N3 fatty acids, which are all commonly recognized elements that can alleviate inflammatory responses. The reason for the higher inflammation scores of these anti-inflammatory nutrients is that patients with CHD have lower intake of these anti-inflammatory nutrients than the global average. The features of high-DII diet of participants with CHD were that they had higher proinflammatory scores especially in various vitamins and fats. Our results may be helpful for the prevention and treatment of CHD. Atherosclerosis is characterized by endothelial damage, inflammation, and vascular smooth muscle cell proliferation as well as deposits of lipids and the extracellular matrix [35]. Considering that Western dietary patterns rich in sugar and fat may increase the risk of CHD, researchers have explored the effect and mechanisms of dietary inflammation on atherosclerosis in an animal model. It has been shown that advanced glycation end products are independently associated with atherosclerosis. A series of complex inflammatory signal transductions include the NF-κB signaling pathway, JAK/STAT signaling pathways, and MAPK signaling pathway. The binding of glycosylation end products (AGEs) and glycosylation end product receptors (RAGE) has been proved to be an important mechanism for the occurrence and development of various chronic diseases [36,37,38]. Moreover, Basta et al. found that AGEs can impair endothelial function by decreasing NO activity and increasing oxidized LDL, ultimately leading to inflammation and the progression of atherosclerosis. High-fat diet alters lipid profiles and aggravates proinflammatory conditions in the vascular walls, which increases CHD risk. On the contrary, the Mediterranean diet is a healthy way of eating and has anti-inflammatory properties [39]. Menotti et al. carried out a multicenter cohort study in Italian rural areas and enrolled 1677 individuals without any heart disease. The incidence of CHD was 28.8% after a 50-year follow-up, and the investigators found that the Mediterranean diet significantly decreased the lifetime incidence of CHD in the fully adjusted Cox regression analysis [40]. Moreover, results of a recent case–control study enrolling 155 heathy individuals and 178 patients with CHD also showed that adherence to the Mediterranean diet can exhibit additive protective effects on CHD. We used multivariable logistic regression analysis to investigate the association of the DII and CHD. Either as continuous variables or as categorical variables grouped by quartiles, high DII was associated with an increased risk of CHD.

RCS, a widely used method, was applied to analyze the nonlinear relationship between variables and outcomes [41,42,43]. RCS is essentially a piecewise polynomial, but it is generally required to be continuous at each piecewise point and second-order derivable so as to ensure the smoothness of the curve. Results of RCS analysis showed that there is a positive association between the DII and CHD risk. It is worth emphasizing that when the DII is greater than 2, the risk of CHD increases rapidly. Therefore, it is very necessary to control the DII in a certain range, and our results may provide new ideas for health policy makers to prevent CHD. A recent large-scale prospective study explored the effects of diet quality on cardiovascular diseases, and the 155,724 participants included came from 21 countries and were followed for approximately 10 years. The primary outcome was a composite of major cardiovascular events, including cardiovascular death, myocardial infarction, stroke, and heart failure. It demonstrated that low diet quality was more strongly associated with cardiovascular diseases in female than in male participants, with an HR of 1.17 (95% CI 1.08–1.26) and 1.07 (0.99–1.15), respectively [44]. Therefore, our study also indicated that quantifying dietary quality with the DII can help prevent CHD in female patients. The DII has different effects on people of different races and different levels of obesity in subgroup analysis, which suggests that we should implement more precise prevention strategies for CHD. In addition to the DII and HEI, there are also some dietary measurements reflecting the quality of diet. The Nutritional Quality Index (NQI) is composed of 35 nutrients, such as protein, fat, cholesterol, dietary fiber, alcohol, and carotene [45]. The intake of each nutrient was compared with the recommended amount to obtain the NQI score. The Diet Balance Index (DBI) is another dietary index, evaluating the intake of 7 kinds of food, including cereals, vegetables and fruits, milk and beans, animal foods, alcohol and condiments, total food types, and drinking water volume. DBI reflects both the abundance and balance of food. The association between various dietary measurements and CHD and how to establish an effective dietary evaluating criteria specific for CHD need to be explored further [46].

There are several advantages and implications of our study. First, it was adequate to provide reliable conclusion and precise statistical power, considering the large-scale sample size included. Second, our study adopted RCS analysis and further demonstrated the nonlinear associations between the DII and CHD, and the trends of RCS curves and cutoff values may provide new evidence for health policy makers. However, several limitations of this study also have to be clarified. First, causal associations could not be determined, considering the research type of cross-sectional study. More prospective studies are needed to determine the exact relationship between the DII and CHD. Second, there may be subjective bias due to the self-reported CHD status and covariates from the NHANES database. Third, there are large ethnic differences in diet, physical activity, genetic variants, lipid metabolism, and susceptibility to cardiovascular disease. Therefore, whether the conclusion in the present study based on US participants could be applicable to other populations needs to be further explored in a future work.

## 5. Conclusions

We retrospectively analyzed 45,036 adults in the US from NHANES and found that the increment in the DII closely was associated with the risk of CHD. The association between the DII and CHD was in a nonlinear pattern, and the increment of the DII had a greater impact on female individuals compared with male individuals. Our results may be helpful to public health policy makers in developing rational approaches to prevent CHD by controlling dietary inflammation, but further studies are still needed.

## Figures and Tables

**Figure 1 nutrients-14-04553-f001:**
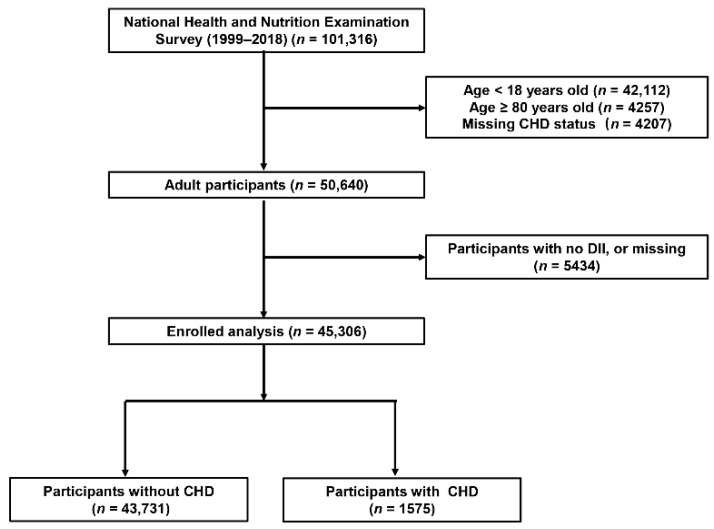
Flowchart of the study population. DII, Dietary Inflammatory Index; CHD, coronary heart disease.

**Figure 2 nutrients-14-04553-f002:**
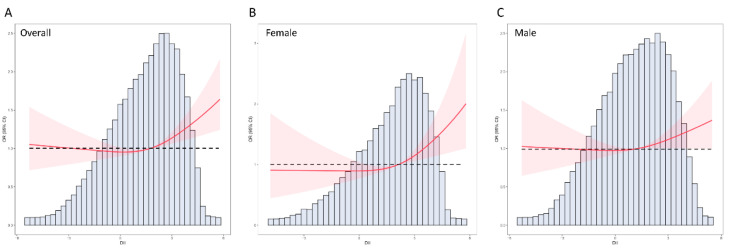
RCS analysis on the association between the DII and the risk of CHD. (**A**) RCS curve of the association between the DII and CHD among all the participants, (**B**) RCS curve of the association between the DII and CHD among female participants, (**C**) RCS curve of the association between the DII and CHD among male participants. RCS: restricted cubic spline; DII: Dietary Inflammatory Index; CHD, coronary heart disease.

**Figure 3 nutrients-14-04553-f003:**
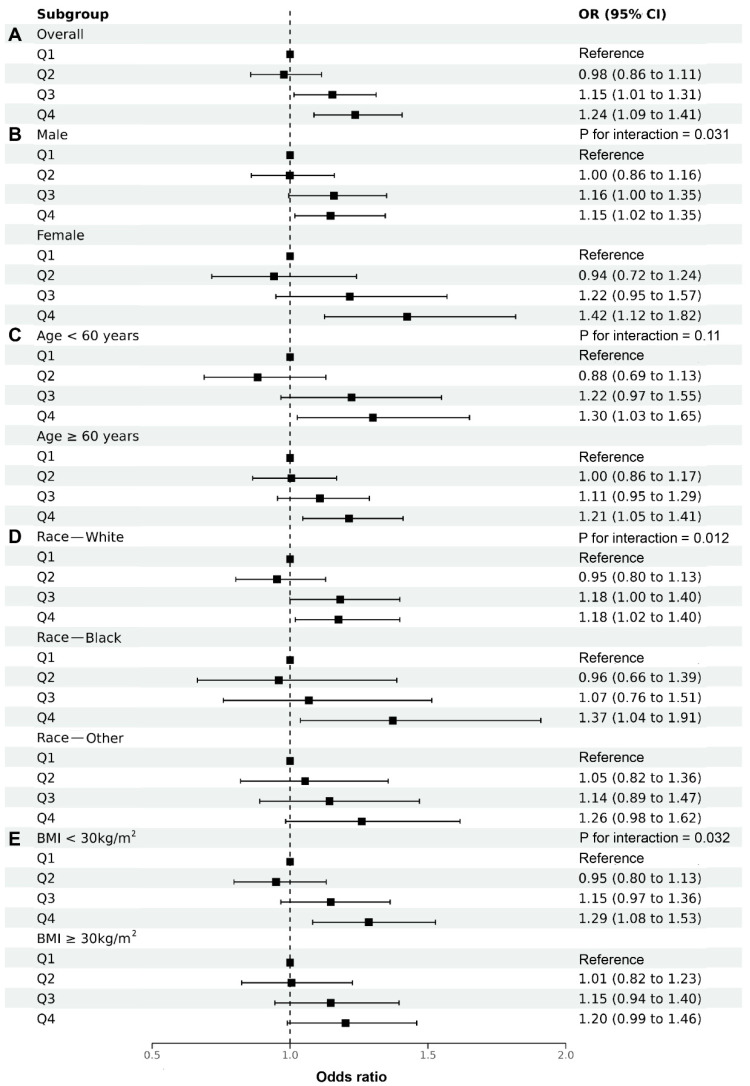
Multivariable logistic regression analysis on the association between the DII and CHD after adjusting for confounding factors of age, sex, race/ethnicity, education levels, smoking, drinking, diabetes, hypertension, and BMI.

**Figure 4 nutrients-14-04553-f004:**
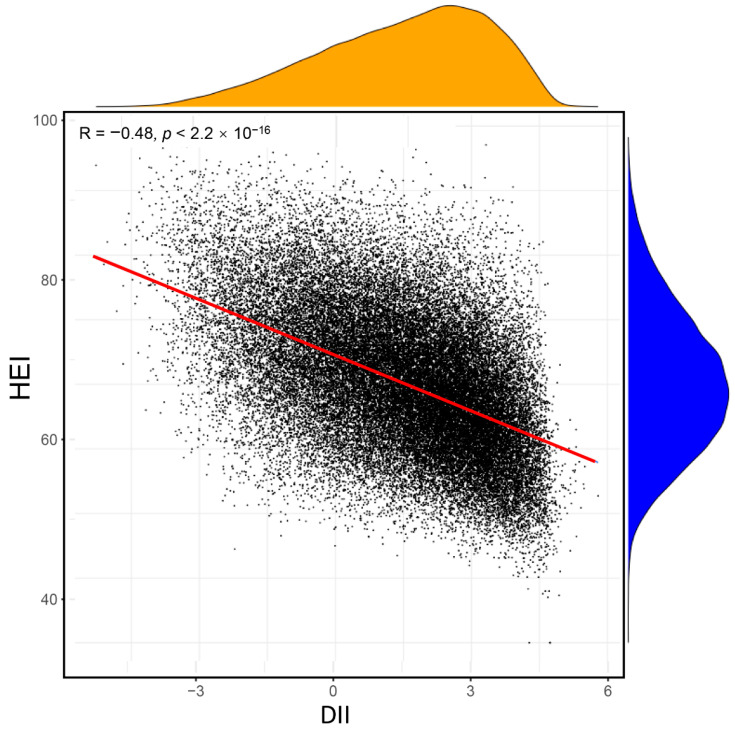
Association between the DII and HEI. DII, Dietary Inflammation Index; HEI, Health Eating Index.

**Table 1 nutrients-14-04553-t001:** Baseline characteristics of all participants.

Variables	Overall (*n* = 45,306)	Non-CHD (*n* = 43,731)	CHD (*n* = 1575)	*p*-Value
Age, years	47.00 (33.00, 61.00)	46.00 (32.00, 61.00)	66.00 (60.00, 73.00)	<0.001 ***
Sex—male, *n* (%)	21,857 (48.2)	20,782 (47.5)	1075 (68.3)	<0.001 ***
Race, *n* (%)				
Non-Hispanic white	19,322 (42.6)	18,400 (42.1)	922 (58.5)	<0.001 ***
Non-Hispanic black	9821 (21.7)	9578 (21.9)	243 (15.4)	
Mexican American	8310 (18.3)	8108 (18.5)	202 (12.8)	
Other Hispanic	3818 (8.4)	3719 (8.5)	99 (6.3)	
Other	4035 (8.9)	3926 (9.0)	109 (6.9)	
Smoking, *n* (%)	9991 (22.1)	9675 (22.1)	316 (20.1)	0.055
Drinking, *n* (%)	12,903 (30.8)	12,266 (30.3)	637 (44.2)	<0.001 ***
Education level, *n* (%)				
Below high school	5069 (11.2)	4807 (11.0)	262 (16.6)	<0.001 ***
High school	17,237 (38.1)	16,594 (38.0)	643 (40.8)	
Above high school	22,964 (50.7)	22,294 (51.0)	670 (42.5)	
BMI, kg/m2	28.03 (24.38, 32.57)	28.00 (24.30, 32.50)	29.28 (25.90, 33.70)	<0.001 ***
SBP, mmHg	120.00 (110.00, 133.00)	120.00 (110.00, 132.00)	128.00 (116.00, 142.00)	<0.001 ***
DBP, mmHg	71.00 (63.00, 78.00)	71.00 (64.00, 78.00)	68.00 (60.00, 76.00)	<0.001 ***
HBP, *n* (%)	18,255 (40.3)	16,984 (38.8)	1271 (80.7)	<0.001 ***
Income > 20,000 USD	31,976 (73.8)	30,975 (74.1)	1001 (66.2)	<0.001 ***
FBG, mmol/L	5.50 (5.11, 6.05)	5.50 (5.08, 6.05)	6.12 (5.50, 7.27)	<0.001 ***
FBI, mmol/L	59.22 (37.80, 96.60)	58.74 (37.50, 95.94)	72.33 (46.35, 119.34)	<0.001 ***
HbA1c, %	5.50 (5.20, 5.80)	5.40 (5.20, 5.80)	5.80 (5.50, 6.70)	<0.001 ***
TG, mmol/L	1.22 (0.84, 1.82)	1.21 (0.84, 1.81)	1.46 (1.00, 2.16)	0.286
TC, mmol/L	4.99 (4.32, 5.74)	5.02 (4.34, 5.74)	4.50 (3.83, 5.33)	<0.001 ***
HDL-C, mmol/L	1.29 (1.06, 1.60)	1.29 (1.08, 1.60)	1.14 (0.98, 1.40)	<0.001 ***
LDL-C, mmol/L	2.92 (2.35, 3.57)	2.95 (2.38, 3.57)	2.43 (1.89, 3.08)	<0.001 ***
Alt, u	21.00 (16.00, 29.00)	21.00 (16.00, 29.00)	21.00 (17.00, 28.00)	0.304 *
Ast, u	22.00 (19.00, 27.00)	22.00 (19.00, 27.00)	23.00 (20.00, 28.00)	0.001 ***
RBC, ×10^9^/L	4.68 (4.35, 5.03)	4.68 (4.35, 5.03)	4.63 (4.27, 4.97)	0.001 ***
WBC, ×10^9^/L	7.00 (5.70, 8.40)	7.00 (5.70, 8.40)	7.10 (5.90, 8.60)	0.015***
PLT, ×10^6^/L	247.00 (209.00, 291.00)	248.00 (210.00, 292.00)	217.00 (180.25, 266.00)	<0.001 ***
Monocyte, ×10^9^/L	0.50 (0.40, 0.70)	0.50 (0.40, 0.60)	0.60 (0.50, 0.70)	<0.001 ***
LY, ×10^9^/L	2.10 (1.70, 2.50)	2.10 (1.70, 2.50)	1.90 (1.50, 2.40)	<0.001 ***
NE, ×10^9^/L	4.00 (3.10, 5.20)	4.00 (3.10, 5.20)	4.20 (3.30, 5.40)	<0.001 ***
Hemoglobin, g/L	14.20 (13.10, 15.20)	14.20 (13.10, 15.20)	14.20 (13.12, 15.10)	0.54
eGFR, ml/min/1.73m^2^	97.44 (81.89, 112.60)	98.17 (82.95, 113.05)	75.77 (60.20, 89.05)	<0.001 ***
DII	1.74 (0.21, 2.94)	1.73 (0.21, 2.93)	1.96 (0.40, 3.13)	<0.001 ***
HEI-2015	49.59 (40.58, 59.28)	49.54 (40.53, 59.22)	51.50 (41.96, 61.19)	<0.001 ***
DM, *n* (%)	7403 (16.9)	6706 (15.8)	697 (44.3)	<0.001 ***
Angina, *n* (%)	1091 (2.4)	577 (1.3)	514 (33.3)	<0.001 ***
Heart attack, *n* (%)	1682 (3.7)	863 (2.0)	819 (52.3)	<0.001 ***
HF, *n* (%)	1189 (2.6)	693 (1.6)	496 (32.1)	<0.001 ***
Stroke, *n* (%)	1462 (3.2)	1212 (2.8)	250 (15.9)	<0.001 ***

Variables are presented as the mean ± standard deviation (SD) (normal distribution), the median (interquartile range) (skewed distribution) or number with percent (categorical). SD, standard deviation; BMI, body mass index; SBP, systolic blood pressure; DBP, diastolic blood pressure; HBP, hypertension; FBG, fasting blood glucose; FBI, fasting blood insulin; HbA1c, glycated hemoglobin; TG, triglycerides; TC, total cholesterol; HDL-C, high-density lipoprotein cholesterol; LDL-C, low-density lipoprotein cholesterol; Alt, alanine transaminase; Ast, glutamic oxalic transaminase; RBC, red blood cells; WBC, white blood cells; PLT, platelets; LY, lymphocytes; NE, neutrophils; eGFR, estimated glomerular filtration rate; DII, Dietary Inflammatory Index; DM, diabetes; HF, heart failure. *** *p*-value < 0.001, * *p*-value < 0.05.

**Table 2 nutrients-14-04553-t002:** Comparison of each component of DII scores between individuals with CHD and individuals without CHD.

Variables	Overall (*n* = 45,023)	Non-CHD (*n* = 43,768)	CHD (*n* = 1255)	*p*-Value
DII	1.74 (0.21, 2.94)	1.73 (0.21, 2.93)	1.96 (0.40, 3.13)	<0.001 ***
Carbohydrate	−0.01 (−0.02, 0.02)	−0.01 (−0.02, 0.02)	−0.01 (−0.02, 0.01)	<0.001 ***
Protein	−0.06 (−0.10, 0.08)	−0.06 (−0.10, 0.08)	−0.08 (−0.10, 0.01)	<0.001 ***
Total fat	0.28 (0.28, 0.28)	0.28 (0.28, 0.28)	0.28 (0.28, 0.28)	<0.001 ***
Fiber	0.01 (−0.23, 0.27)	0.01 (−0.23, 0.27)	−0.10 (−0.26, 0.20)	<0.001 ***
Cholesterol	0.00 (−0.01, 0.01)	0.00 (−0.01, 0.01)	0.00 (−0.01, 0.01)	<0.001 ***
Saturated fat	0.43 (−0.25, 0.63)	0.43 (−0.25, 0.63)	0.45 (−0.17, 0.64)	0.004 **
MUFA	−0.08 (−0.11, 0.11)	−0.07 (−0.11, 0.11)	−0.10 (−0.11, 0.10)	<0.001 ***
PUFA	−0.20 (−0.34, 0.19)	−0.20 (−0.34, 0.19)	−0.28 (−0.36, 0.02)	<0.001 ***
Niacin	−0.01 (−0.05, 0.03)	−0.01 (−0.05, 0.03)	−0.01 (−0.05, 0.03)	<0.001 ***
Vitamin A	−0.10 (−0.33, 0.25)	−0.10 (−0.33, 0.25)	0.04 (−0.30, 0.28)	<0.001 ***
Thiamin	0.27 (0.12, 0.34)	0.27 (0.12, 0.34)	0.27 (0.14, 0.34)	<0.001 ***
Riboflavin	0.03 (−0.03, 0.07)	0.03 (−0.03, 0.07)	0.04 (−0.03, 0.07)	0.021 *
Vitamin B6	−0.08 (−0.30, 0.13)	−0.09 (−0.30, 0.13)	−0.03 (−0.27, 0.16)	<0.001 ***
Vitamin B12	0.36 (−0.02, 0.41)	0.36 (−0.02, 0.41)	0.37 (0.04, 0.41)	0.045 *
Vitamin C	0.44 (0.30, 36.95)	0.44 (0.30, 36.81)	0.44 (0.32, 41.34)	<0.001 ***
Vitamin D	0.37 (−0.23, 0.42)	0.37 (−0.23, 0.42)	0.40 (−0.07, 0.42)	<0.001 ***
Vitamin E	0.00 (−0.02, 0.03)	0.00 (−0.02, 0.03)	−0.01 (−0.03, 0.02)	0.003 **
Iron	0.01 (−0.30, 0.27)	0.01 (−0.30, 0.27)	0.10 (−0.27, 0.28)	<0.001 ***
Magnesium	0.13 (−0.14, 0.30)	0.12 (−0.14, 0.30)	0.17 (−0.08, 0.32)	<0.001 ***
Zinc	−0.15 (−0.19, −0.01)	−0.15 (−0.19, −0.01)	−0.13 (−0.19, 0.03)	<0.001 ***
Selenium	0.18 (0.11, 0.19)	0.18 (0.11, 0.19)	0.19 (0.13, 0.19)	0.012 *
Folic acid	−0.04 (−0.08, 0.03)	−0.04 (−0.08, 0.03)	−0.05 (−0.08, 0.02)	0.001 **
β-Carotene	0.07 (−0.08, 0.16)	0.07 (−0.08, 0.16)	0.09 (−0.04, 0.17)	<0.001 ***
Caffeine	0.54 (0.38, 0.56)	0.54 (0.38, 0.56)	0.54 (0.40, 0.56)	0.081
Alcohol	0.08 (0.08, 0.08)	0.08 (0.08, 0.08)	0.08 (0.08, 0.08)	<0.001 ***
Energy	−0.04 (−0.17, 0.16)	−0.04 (−0.17, 0.17)	−0.12 (−0.18, 0.07)	<0.001 ***
n3 fatty acids	0.29 (0.28, 0.30)	0.29 (0.28, 0.30)	0.29 (0.28, 0.30)	<0.001 ***
n6 fatty acids	−0.07 (−0.14, 0.02)	−0.07 (−0.14, 0.02)	−0.04 (−0.12, 0.04)	<0.001 ***

Variables are presented as the mean ± standard deviation (SD) (normal distribution), the median (interquartile range) (skewed distribution). DII, Dietary Inflammatory Index; CHD, coronary heart disease; MUFA, monounsaturated fatty acids; PUFA, polyunsaturated fatty acids. *** *p*-value < 0.001, ** *p*-value < 0.01, * *p*-value < 0.05.

**Table 3 nutrients-14-04553-t003:** Logistic regression analysis on the association between the DII and CHD.

	Nonadjusted Model		Model I		Model II	
	OR (95% CI)	*p*-Value	OR (95% CI)	*p*-Value	OR (95% CI)	*p*-Value
DII	1.09 (1.05, 1.14)	<0.001 ***	1.17 (1.11, 1.22]	<0.001 ***	1.08 (1.03, 1.14))	0.005
Q1	Reference	-	Reference	-	Reference	-
Q2	0.99 (0.87, 1.12)	0.849	1.05 (0.93, 1.2)	0.512	0.98 (0.86, 1.11)	0.776
Q3	1.14 (1.01, 1.29)	0.067 *	1.3 (1.15, 1.48)	<0.001 ***	1.15 (1.01, 1.31)	0.041 *
Q4	1.26 (1.12, 1.42)	0.001 **	1.48 (1.31, 1.68)	<0.001 ***	1.24 (1.09, 1.41)	0.007 **

Data are presented as OR (95% CI). Model I adjusted for age, sex, and race/ethnicity. Model II adjusted for age, sex, race/ethnicity, education levels, smoking, drinking, diabetes, hypertension, and BMI. OR, odds ratio; CI, confidence interval; BMI, body mass index; DII, Dietary Inflammation Index; Q1, first quartile; Q2, second quartile; Q3, third quartile; Q4, fourth quartile. *** *p*-value < 0.001, ** *p*-value < 0.01, * *p*-value < 0.05.

**Table 4 nutrients-14-04553-t004:** Logistic regression analysis including both the DII and HEI.

	Nonadjusted Model		Model I		Model II	
	OR (95% CI)	*p*-Value	OR (95% CI)	*p*-Value	OR (95% CI)	*p*-Value
DII	1.1 (1.07, 1.13)	<0.001 ***	1.09 (1.06, 1.12)	<0.001 ***	1.05 (1.02, 1.08)	0.005 **
Q1	Reference	-	Reference	-	Reference	-
Q2	1.07 (0.94, 1.21)	0.39	1.07 (0.93, 1.22)	0.424	1 (0.87, 1.14)	0.028 *
Q3	1.33 (1.17, 1.51)	<0.001 ***	1.33 (1.16, 1.52)	<0.001 ***	1.2 (1.05, 1.37)	0.003 **
Q4	1.56 (1.37, 1.78)	<0.001 ***	1.5 (1.3, 1.74)	<0.001 ***	1.3 (1.12, 1.5)	<0.001 ***
HEI	1.01 (1.00, 1.02)	<0.001 ***	1.00 (0.99, 1.00)	0.977	1 (1.00, 1.01)	0.418
Q1	Reference	-	Reference	-	Reference	-
Q2	1.15 (1.00, 1.31)	0.07	0.98 (0.86, 1.12)	0.78	1 (0.88, 1.15)	0.975
Q3	1.31 (1.15, 1.49)	<0.001 ***	0.98 (0.85, 1.12)	0.771	1.02 (0.89, 1.18)	0.777
Q4	1.65 (1.45, 1.89)	<0.001 ***	1.04 (0.9, 1.2)	0.666	1.11 (0.96, 1.29)	0.22

Data are presented as OR (95% CI). Model I adjusted for age, sex, and race/ethnicity. Model II adjusted for age, sex, race/ethnicity, education levels, smoking, drinking, diabetes, hypertension, and BMI. OR, odds ratio; CI, confidence interval; BMI, body mass index; DII, Dietary Inflammation Index; HEI, Healthy Eating Index; Q1, first quartile; Q2, second quartile; Q3, third quartile; Q4, fourth quartile. *** *p*-value < 0.001, ** *p*-value < 0.01, * *p*-value < 0.05.

## Data Availability

Publicly available datasets were analyzed in this study. All the raw data used in this study are derived from the public NHANES data portal (https://wwwn.cdc.gov/nchs/nhanes/analyticguidelines.aspx (accessed on 1 August 2022)).

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
