# Peer review of "Dietary Inflammatory Index and Its Association with the Prevalence of Coronary Heart Disease among 45,306 US Adults"

_nutrients, 2022, doi:10.3390/nu14214553_

Round 1

Reviewer 1 Report

The aim of the current study was to explore the link between dietary inflammatory index (DII) and CHD in participants from the National Health and Nutrition Examination Survey (NHANES).

The study has good potential but there are issues to be addressed.

1. Within the sample, patients who reported having had a CHD were selected. It is possible that this type of patient selection is inaccurate and full of BIAS. The authors should clarify.

2. Throughout the paper, the characteristics of the DII diet are not effectively described. Some rather general correlations are illustrated in Table 2. What are the main features of the DII diet evaluated in this group of patients?

3. The text seems to be in partial form, there are choreographies and some figures do not have captions. 

4. Figure 3 is the most important as it highlights the associations between CHD and DII adjusted for confounding factors. It is not very clear and even in the text there are few references. The authors should improve this part. 

5. The authors correctly point out in the discussion that the Mediterranean diet and plant-based diets are the most effective in reducing the inflammatory effect of obesity. Some recent work erroneously emphasises the ketogenic diet as anti-inflammatory. Its partial anti-inflammatory effect is due to the momentary and rapid weight reduction of these low-carbohydrate diets. The authors should investigate these aspects further. 

6. Beware of plagiarism between methods (see attached file). 

Author Response

A point-by-point response to Reviewer 1

We sincerely thank Editor and Reviewers for your very constructive comments. We have made changes according to your suggestions. Our point-by-point responses are listed below:

Reviewer 1:

The aim of the current study was to explore the link between dietary inflammatory index (DII) and CHD in participants from the National Health and Nutrition Examination Survey (NHANES). The study has good potential but there are issues to be addressed.

We sincerely thank you for your positive comments. We have revised our manuscript as you suggested.

Within the sample, patients who reported having had a CHD were selected. It is possible that this type of patient selection is inaccurate and full of BIAS. The authors should clarify.

This is an insightful comment! We totally agree with you that bias may existed considering the NHANES database is a large-scale cross-sectional study, some of the outcomes and covariates are collected through questionnaire. Though, the process of collecting information and the answers of participants in the questionnaire were evaluated by the experienced NHANES investigator, some of the results could still be inaccurate. Therefore, there may exist experienced subjective bias in the present study. We have pointed this issue out in the Discussion section (Page: 12; Line: 532-533).

Throughout the paper, the characteristics of the DII diet are not effectively described. Some rather general correlations are illustrated in Table 2. What are the main features of the DII diet evaluated in this group of patients?

We sincerely thank you for this helpful comment! We have revised our manuscript and strengthen the features of high-DII diet in the Discussion section. Just as you pointed out that in participants with CHD had higher inflammatory scores in dietary-fibre, MUFA, PUFA, niacin, vitamin A, thiamin, riboflavin, vitamin B6, vitamin C, vitamin D, vita-min E, magnesium, zinc, selenium, folic acid, β-carotene, alcohol, N6 fatty acids and N3 fatty acids, which were all commonly recognized elements that could lead to (such as alcohol) or alleviate (dietary-fibre, riboflavin, vitamin B6) inflammatory responses. We observed in the present study participants with CHD had higher DII scores, especially in vitamins and various fats. Actually, there was already a large number animal researches or clinical trials, indicating ingredients in our daily diet have an impact on the systemic inflammation level in the human body. DII, a novel index, also derived from literatures, has already been used to accurately evaluate the effect of daily diet habit on systematic inflammation. Thanks again for your very helpful comment and we have addressed the features of high-DII diet of patients with CHD in the Discussion section (Page: 11; Line: 443-450).

The text seems to be in partial form, there are choreographies and some figures do not have captions.

Sorry for this mistake! Thank you for your kindly reminding! We have corrected our manuscript as you suggested.

Figure 3 is the most important as it highlights the associations between CHD and DII adjusted for confounding factors. It is not very clear and even in the text there are few references. The authors should improve this part.

Excellent suggestion! Figure 3 is definitely a very important part of this study, as you suggested, we have revised our manuscript and explained this result in detail in the Results part (Page: 7; Line: 310-320).

The authors correctly point out in the discussion that the Mediterranean diet and plant-based diets are the most effective in reducing the inflammatory effect of obesity. Some recent work erroneously emphasises the ketogenic diet as anti-inflammatory. Its partial anti-inflammatory effect is due to the momentary and rapid weight reduction of these low-carbohydrate diets. The authors should investigate these aspects further.

Thank you for your insightful comment! Our views on the ketogenic diet are in line with yours. Actually, in the present study, we can also draw the conclusion that ketogenic diet characterized by high fat, very low carbohydrate and moderate protein can eventually lead to the elevated DII. For most of people, the risks of a ketogenic diet clearly outweigh the benefits, we have strengthened it in the Discussion section. Our group was also focusing on the ketogenic diet, in our future work, we hope to investigate the accurate association between ketogenic diet and inflammation and the association between ketogenic diet and CHD (Page: 10; Line: 388-393).

Beware of plagiarism between methods (see attached file).

Thank you very much for your helpful comment and kindly reminding. We have revised the manuscript to ensure that the proportion of overlapping words meets the requirements.

Thank you again for your time and very helpful comments!

Reviewer 2 Report

Wu review

Comments and suggestions:

1.       The first paragraph of the Introduction does not give an accurate and current summary of atherosclerosis and atherosclerotic disease. It fails to discuss soft, vulnerable plaque, or the acute thrombosis that typically precipitates vascular events even when the underlying plaque is far from being occlusive up to that point. This section could be improved by relying on recent reviews, such as https://doi.org/10.1016/j.cell.2022.04.004.

2.       The second paragraph of the Introduction needs to address the kinds of dietary fat linked to CHD and to inflammation. In addition, unless I am mistaken, vitamin E is expressly NOT recommended for treatment or prevention of heart disease since RCT evidence have proven it ineffective.

3.       Please define in greater detail how CHD was operationalized in this study. What questions exactly from the NHANES were considered to represent CHD?  Why not include other or all medical diagnoses that constitute clinical manifestations of atherosclerotic disease?

4.       The validation of the DII is asserted without presentation of the evidence. Please add such to the Methods paragraph where the DII is described.

5.       This is a cross-sectional design and that does significantly handicap any conclusions that can be drawn. This limitation is not raised until the final section of the Discussion and is not actually addressed. Please explain early in the Discussion how we should view the findings in light of this weakness. In this, readers would benefit from understanding what additional would be learned with a prospective design.

6.       I note that very small proportion (<4%) of participants who had a history of CHD. Moreover, those with and without CHD differ in age by an average of 20 years. So, I have to think the results are confounded by age-cohort-generation effects. The data lend themselves nicely to a case-control design with matching an age, gender, and race. I recommend this as a significantly stronger design for the investigators to consider.

7.       The DII is rather new and must be discriminated from the well-established Healthy Diet Index (Healthy Eating Index | Food and Nutrition Service (usda.gov)).  We clearly need to understand the degree of correlation between the HEI and the DII. In addition, comparing the two would be a basis for determining the unique value of the DII (ie, discriminate validity). Therefore, I urge these authors to also calculate the HEI and present the correlation between the two indices, and how they perform in models including both.

8.       In Table 2, the average BP in the two groups looks plausible but conflict with immediately below where the CHD subjects have HTN rarely whereas non-CHD subjects have a HTN prevalence of 60%(!).

9.       The table with subgroup analyses could be improved by including the P for interaction for each comparison.

10.   I do not view the presented “Sensitivity analysis” as actually constituting a sensitivity analysis. I suggest dropping that section or replacing it with a more traditional sensitivity analysis.

11.   Discussion would be strengthened by a) adding discussion of other dietary measures linked prospectively to CHD, and b) cutting by half the paragraph on RCS analysis.

Author Response

A point-by-point response to Reviewer 2

We sincerely thank Editor and Reviewers for your very constructive comments. We have made changes according to your suggestions. Our point-by-point responses are listed below:

Reviewer 2

The first paragraph of the Introduction does not give an accurate and current summary of atherosclerosis and atherosclerotic disease. It fails to discuss soft, vulnerable plaque, or the acute thrombosis that typically precipitates vascular events even when the underlying plaque is far from being occlusive up to that point. This section could be improved by relying on recent reviews, such as https://doi.org/10.1016/j.cell.2022.04.004.

Thank you for your very constructive suggestion! we have revised our manuscript and rewrite the first paragraph to summarize atherosclerosis and atherosclerotic disease more accurately. In addition, we have added the introduction of soft, vulnerable plaque, and the acute thrombosis (Page: 1; Line: 37-39).

The second paragraph of the Introduction needs to address the kinds of dietary fat linked to CHD and to inflammation. In addition, unless I am mistaken, vitamin E is expressly NOT recommended for treatment or prevention of heart disease since RCT evidence have proven it ineffective.

We sincerely thank you for your helpful suggestion! We have revised this paragraph and addressed the kinds of dietary fat linked to CHD and to inflammation. Thank you for your reminding, we have also deleted the vitamin E part (Page: 2; Line: 63-70).

Please define in greater detail how CHD was operationalized in this study. What questions exactly from the NHANES were considered to represent CHD? Why not include other or all medical diagnoses that constitute clinical manifestations of atherosclerotic disease?

This is an insightful comment! In NHANES questionnaire, the question is “has a doctor or other health professional ever told you that you had coronary heart disease”. Because participants included in the NHANES did not have coronary angiography results, CHD could only be studied with a questionnaire. We cannot infer whether a patient has coronary heart disease through the use of some special drugs, such as aspirin and clopidogrel, because these drugs are not the only specific drugs for coronary heart disease, not only for coronary heart disease, peripheral artery disease and other diseases can be used. Other previous published studies focused on risk factors and the occurrence of CHD also used the questionnaire as we did (such as DOI: 10.1016/j.numecd.2017.07.007). There may be subjective bias due to the self-reported CHD status from NHANES database, we have strengthened this issue in the Discussion section (Page: 10; Line: 532-533).

The validation of the DII is asserted without presentation of the evidence. Please add such to the Methods paragraph where the DII is described.

Excellent and very useful suggestion! As you suggested, we have added evidence that validated the effectiveness of DII in evaluating systemic inflammation into the Methods part (Page: 3; Line: 143-146).

This is a cross-sectional design and that does significantly handicap any conclusions that can be drawn. This limitation is not raised until the final section of the Discussion and is not actually addressed. Please explain early in the Discussion how we should view the findings in light of this weakness. In this, readers would benefit from understanding what additional would be learned with a prospective design.

Thanks for this good suggestion! We totally agree with you that more prospective studies are needed to further explore the association between DII and CHD. Considering this study is a cross-sectional study, we can only draw correlation conclusions. We have explained this issue in the very beginning of Discussion section (Page: 10; Line: 422-423).

I note that very small proportion (<4%) of participants who had a history of CHD. Moreover, those with and without CHD differ in age by an average of 20 years. So, I have to think the results are confounded by age-cohort-generation effects. The data lend themselves nicely to a case-control design with matching an age, gender, and race. I recommend this as a significantly stronger design for the investigators to consider.

Excellent comment! Actually, our group recently focusing on the impact of dietary inflammation on CHD, we plan to conduct more research about DII in our future work. We highly agree with you the data lend themselves nicely to a case-control design with matching an age, gender, and race using PSM, it is a good idea. Besides, another limitation of the present study is that the NHANES is more like a large-scale cross-sectional survey, it has no follows-ups data except for the all-cause mortality and cardiovascular cause mortality. In our future work, we hope to further explore the impact of DII and the prevalence of CHD, MI, and stroke, and the prognosis based on our own data.

The DII is rather new and must be discriminated from the well-established Healthy Diet Index (Healthy Eating Index | Food and Nutrition Service (usda.gov)). We clearly need to understand the degree of correlation between the HEI and the DII. In addition, comparing the two would be a basis for determining the unique value of the DII (ie, discriminate validity). Therefore, I urge these authors to also calculate the HEI and present the correlation between the two indices, and how they perform in models including both.

Good idea! Thank you so much for your helpful suggestion, which can improve the quality of our manuscript! DII is actually a novel index and it is necessary to compare the effectiveness of DII with HEI. Moreover, investigating the correlation between DII and HEI would be very interesting and meaningful. As you suggested, we also calculated the HEI according to the 2015 version. The association between DII and HEI was shown in Figure 6, we found that DII negatively associated with HEI (R = -0.48, P < 0.001). However, results of logistic regression analysis included both DII and HEI demonstrated that DII are closely associated with CHD but not HEI after adjusting for cofounders. Therefore, for prevention of CHD, individuals should pay more attention to DII.

In Table 2, the average BP in the two groups looks plausible but conflict with immediately below where the CHD subjects have HTN rarely whereas non-CHD subjects have a HTN prevalence of 60%(!).

Sorry for this mistake! Thanks a lot for your kindly reminding! We miscalculated the percentage of participants with hypertension as the percentage of participants without hypertension, we have corrected this mistake.

The table with subgroup analyses could be improved by including the P for interaction for each comparison.

Very useful suggestion! We have revised the Figure 3 and added the P for interaction for each comparison in the subgroup analysis.

I do not view the presented “Sensitivity analysis” as actually constituting a sensitivity analysis. I suggest dropping that section or replacing it with a more traditional sensitivity analysis.

Thank you for your helpful suggestion! we have deleted the “Sensitivity analysis” section as you suggested.

Discussion would be strengthened by a) adding discussion of other dietary measures linked prospectively to CHD, and b) cutting by half the paragraph on RCS analysis.

Thank you for your constructive comment! We have added dietary measures linked prospectively to CHD in the discussion and cutting half of the paragraph on RCS analysis (Page: 12; Line: 622-630).

Thank you again for your time and very helpful comments!

Round 2

Reviewer 1 Report

Authors have well adressed all my concerns about this study

Author Response

Thank you again for your time and very constructive comments!